# An Urgent Call for Concussion Incidence Measures in Para Sport for Athletes with Vision Impairment: A Narrative Review

**DOI:** 10.3390/healthcare12050525

**Published:** 2024-02-22

**Authors:** Juliette I. Teodoro, Elizabeth L. Irving, Jane D. Blaine, Kristine N. Dalton

**Affiliations:** 1Vision and Motor Performance Lab, School of Optometry and Vision Science, Faculty of Science, University of Waterloo, Waterloo, ON N2L 3G1, Canada; jteodoro@uwaterloo.ca; 2School of Optometry and Vision Science, Faculty of Science, University of Waterloo, Waterloo, ON N2L 3G1, Canada; elizabeth.irving@uwaterloo.ca; 3British Columbia Blind Sports and Recreation Association, Burnaby, BC V5H 4K7, Canada; jane@bcblindsports.bc.ca

**Keywords:** sport-related concussion, para athlete, low vision, blindness, injury epidemiology, injury mechanisms, injury prevention, supporting structures

## Abstract

Concussion in para athletes with vision impairment (VI) is poorly understood. Recently published studies have suggested that athletes with VI may be more likely to sustain sport-related concussions compared to non-disabled athletes and athletes with other impairment types. There is a critical need for objective concussion incidence measures to determine concussion injury rates and risks more accurately. The aim of this review was to examine the limited available evidence of concussion incidence rates across six different para sports for athletes with VI and encourage the future collection of concussion incidence data and the adoption of injury prevention strategies in VI para sport. A literature search was conducted using four unique databases, which formed the basis of this narrative review. Injury prevention strategies such as modifying sport rules, introducing protective equipment, and incorporating additional safety measures into the field of play have been introduced sporadically, but the effectiveness of most strategies remains unknown. More prospective, sport-specific research examining mechanisms of injury and risk factors for concussion injuries in athletes with VI in both training and competition is needed. This research will help inform the development of targeted injury prevention strategies to reduce the likelihood of concussion for athletes with VI.

## 1. Introduction

Through engagement in organized sport, both recreational and competitive athletes can experience improvements in their physical health, mental health, and overall life satisfaction [1]. Sport participation has been shown to not only reduce obesity and improve overall physical health but also appears to contribute to higher life satisfaction in the general population by four times the size of the coefficient on unemployment status [2]. However, participation in sports also increases one’s risk of sustaining various injuries. One common sport-related injury is sport-related concussion, which is a mild traumatic brain injury caused by a direct or indirect impact to the head, face, or neck that results in disturbances in brain function [3]. While disturbances in brain function following a concussion are usually transient in nature [3,4], it is estimated that up to 30% of adults who sustain sport-related concussions will experience persisting post-concussion symptoms [5,6]. The global incidence of concussions from all causes is estimated to be approximately 42 million per year [7]. Sport-related concussions are thought to account for approximately 20%, or 8.4 million, of these injuries per year [8].

Concussion among non-disabled athletes is comparatively well studied in relation to concussion in para athletes—an International Paralympic Committee (IPC) term used to describe pro and amateur athletes with a physical, intellectual, or vision impairment (VI) [9,10]. A literature search conducted using EMBASE, Ovid Medline, PubMed, and Scopus databases revealed that as of 13 December 2023, there were a total of 54 results for studies related to concussion in para, adaptive, or disability sport, compared to 7960 results on PubMed alone for concussion in sport more generally.

Throughout this review, we use terminology consistent with the IPC’s Guide to Para and IPC Terminology, developed from the UN Convention on the Rights of Persons with Disabilities [10]. The terms Paralympic, Paralympics, and Paralympian can only be used with reference to the Paralympic Games, while the term ‘para’ can be used in sport contexts outside of the Paralympic Games, at both the pro and amateur levels [10]. The term ‘para sport’ is defined as any sport in which athletes with a disability or impairment participate and whereby the International Federation has been recognized by the IPC [10]. It is important to note that not all para sports are included in the Paralympic Games sport program [10]. It is also worth noting that some para sports are exclusive to one impairment group (e.g., Goalball is exclusive for athletes with VI), while others are practiced by more than one impairment group (e.g., athletes with VI, physical impairments, and intellectual impairments compete in Para Swimming).

In para sport, VI is defined as a reduction (low vision) or absence of vision (blindness) caused by damage to the eye structure, optical nerves or optical pathways, or visual cortex of the brain [11]. A number of underlying health conditions may lead to VI, two examples of which are retinitis pigmentosa and diabetic retinopathy [11]. Athletes with VI generally compete in three sport classes, depending on the relative severity of their impairment: B1, visual acuity worse than 2.6 LogMAR or no light perception (i.e., total blindness); B2, visual acuity ranging from 1.5 to 2.6 LogMAR or visual field constricted to a diameter of less than 10 degrees; and B3, visual acuity ranging from 1.0 to 1.4 LogMAR or visual field constricted to a diameter of less than 40 degrees [12].

There has been considerable effort over the last two decades to address the lack of injury incidence data in para sports. The IPC has conducted injury surveillance studies at every Paralympic Winter Games since 2002 and at every Paralympic Games since 2012 [9]. London 2012 saw the introduction of the IPC’s web-based injury and illness surveillance system (WEB-IISS), which allowed for the collection of more comprehensive injury data than had previously been possible [13]. The WEB-IISS also enhanced injury reporting compliance by team medical staff and facilitated the collection of exposure data, thereby allowing for more accurate calculations of rates of illness and injury [13].

While there has been progress in the epidemiology of injuries and illnesses in general in para sport, our knowledge of the true burden of concussion in para sport is largely unknown. Though the WEB-IISS was introduced at the 2012 Paralympic Games, questions specifically related to concussion were not added to the web-based injury surveillance system until the Rio 2016 Paralympic Games [9]. Even with the inclusion of specific questions regarding concussion, the authors of the 2016 Paralympic Games prospective cohort study stated concussions may have been underreported in the study population of para athletes [14]. At the Pyeongchang 2018 Paralympic Winter Games, a total of four concussions were reported among 29 injuries to the head, face, or neck, including two in Para Alpine Skiing and two in Para Ice Hockey [15]. There was no further elaboration in terms of which athlete impairment type (i.e., limb deficiency, impaired muscle power, VI, etc.) these concussions occurred in. More recently, at the Tokyo 2020 Paralympic Games, a total of nine concussion-related injuries were reported, including three in Para Judo, two in Para Road Cycling, two in Para Taekwondo, one in Para Swimming, and one concussion that was non-sport-related [16]. It was also noted that four out of nine (44%) of these concussions occurred in athletes with VI [16]. An estimated total of 719 athletes with VI (estimates of athlete numbers per impairment type were made by the authors based on publicly available data from the Tokyo 2020 Paralympic Games) participated in the Tokyo 2020 Paralympic Games, meaning approximately 0.56% of athletes with VI sustained a concussion injury. In contrast, an estimated 3523 athletes with physical impairment competed in the Tokyo 2020 Games, suggesting approximately 0.14% of athletes with physical impairment sustained a concussion injury. The statistic above suggests athletes with VI may be up to four times more likely to sustain a concussion injury compared to athletes with physical impairment. While it is encouraging to see improved concussion reporting from one Paralympic Games to the next, further understanding of the mechanisms of injury involved in the sustainment of concussions in each para sport is needed to inform injury prevention strategies going forward.

One of the first para-sport-specific studies to measure concussion incidence rates was conducted at the Fifth World Para Taekwondo Championships in 2014 [17]. Researchers found the total concussion incidence among male and female athletes with physical impairments was 6.6 per 1000 athletic exposures and 3.2 per 1000 min-exposures [17]. Two other potential concussion injuries were observed during competition; however, due to limitations in injury reporting and conservative definitions of concussive states, both alleged concussions were not included in the results analysis [17]. This finding suggests that concussion incidence rates may be higher than those reported in the literature.

A recent study by Lexell and colleagues used data collected from the Sports-Related Injuries and Illnesses in Paralympic Sport Study (SRIIPSS) and assessed the incidence proportion and incidence rate of sport-related concussion among elite Swedish para athletes of a variety of impairment types [18]. The SRIIPSS, a 52-week prospective longitudinal cohort study, was the first of its kind to prospectively assess the epidemiology of sport-related injuries and illnesses in para sports over a longer period of time [19]. The authors found that athletes with VI reported a significantly higher incidence proportion of sport-related concussion and a significantly higher incidence rate of sport-related concussion than athletes of other impairment types (i.e., athletes with physical impairments and athletes with intellectual impairments) [18]. In addition, researchers found almost two-thirds (62%) of all sport-related concussions reported over a period of 52 weeks occurred among athletes participating in VI para sports, even though athletes with VI only accounted for approximately 21% of all athletes who participated in the study (i.e., participants included 78 athletes with physical impairment, 22 athletes with VI, and 6 athletes with intellectual impairment) [18]. Finally, Lexell and colleagues found that all concussions reported in this study in VI para sports occurred in Goalball, Para Judo, and Para Swimming [18]. These results suggest once again that athletes with VI may be more likely to sustain a sport-related concussion compared to athletes with other impairment types. More prospective, sport-specific concussion injury incidence studies are clearly needed to determine injury rates and risks more accurately. It would also be useful to understand the injury mechanisms and risk factors that contribute to the seemingly higher concussion risk in VI para sports so that targeted preventive measures can be taken to minimize or prevent concussion in sports for athletes with VI.

To address the limited available concussion injury incidence data specific to impairment type and sport, members of the IPC Medical Committee created a ‘best estimate’ risk assessment for para sports based on their experience [9]. Impairment type, speed, collision potential, and whether protective head gear is worn in each sport were also considered in determining the risk assessment [9]. A concussion risk rating was then assigned to each summer and winter sport, specific to impairment type, ranging from a score of ‘1′ for low risk to a score of ‘5’ for high risk [9]. The VI sports Blind Football, Goalball, Para Cycling, Para Alpine Skiing, and Para Triathlon (Bike) were all rated a ‘3’ or higher, putting these sports at an estimated moderate to very high risk of concussion injury in comparison to other VI para sports [9]. As such, Blind Football, Goalball, Para Cycling, and Para Alpine Skiing were included in this review. Para Judo and Para Swimming were also included in this review, as there is some evidence to suggest athletes with VI are at an elevated risk of concussion in these sports as well.

The aim of this review is to examine the available evidence of concussion incidence rates across different para sports for athletes with VI and encourage the future collection of concussion incidence data and the adoption of injury prevention strategies in VI para sport. Mechanisms of injury and risk factors specific to each VI para sport are hypothesized given current available data on concussion incidence rates and exposure risk. Where available, current injury prevention strategies are discussed. Recommendations for future concussion incidence reporting and considerations for future injury prevention strategies are also provided.

## 2. Sport-Specific Concussion Incidence

### 2.1. Blind Football

Blind Football is a para sport played exclusively by athletes with VI. All outfield players are classified as completely blind (B1) and are required to wear eye shades to ensure fair competition [12,20]. Goalkeepers are partially sighted and can be classified as either B2 or B3 [12,20]. Goalkeepers do not wear eye shades during play.

The IPC Medical Committee rated Blind Football a ‘4′ out of a possible ‘5’ in terms of a best estimate of concussion risk, putting the sport at a perceived moderate-to-high risk for concussion compared to other para sports [9]. Although the impact speed in Blind Football is much slower than downhill Para Alpine, for instance, the collision potential remains quite high considering all outfield players have no light perception on the field of play and are unable to see incoming objects or other athletes [9]. Athletes with VI must rely substantially on their hearing ability to know where the ball, equipped with a sound system inside it, is and where other players around them are on the field of play [21]. As outfield players are completely blind, the athlete’s ability to brace for or block impact to the head is reduced [21]. Observations from Blind Football indicate that athletes with VI tend to play with a more anterior posture compared to their non-disabled counterparts, which could expose athletes with VI to an even greater risk of head-to-head collision [21,22]. These risk factors are coupled with the fact that no head protection is worn in Blind Football. It is suspected that head protection would reduce the likelihood of concussions occurring in the sport [23]. Results of a study by the International Blind Sports Federation (IBSA) testing the effectiveness of head protection are pending [23].

The first group to independently investigate the nature and incidence of sport-related injuries in Blind Football over a period of five years was Magno e Silva and colleagues [22]. They recruited a total of 13 Brazilian male Blind Football athletes, and these athletes competed in five international competitions between 2004 and 2008 [22]. A standardized injury report form was used to document all athlete injuries that occurred during all major international competitions [22]. The report form documented the athlete’s injured body part, mechanism of injury, and diagnosis of injury [22]. The study found that 11 out of 13 athletes suffered some form of injury over the course of the five international competitions, which represented an injury prevalence of 84.6% [22]. Injuries to the head represented the body region with the second highest injury prevalence (i.e., 8.6% of all injuries) [22]. The authors of the study recognized that head injuries are of tremendous concern to the healthcare team as they often lead to concussions. The study did not document the head injuries more specifically as concussions [22]. In addition to documenting the anterior posture Blind Football athletes tend to adopt while running, Magno e Silva et al. also found some athletes had developed homemade eye shades made of an absorbent material with padding on the front and parietal zones of the head to decrease the possibility of severe injury to the head and face regions [22]. Nevertheless, this form of protective equipment is optional, and even at the most recent Tokyo 2020 Paralympic Games, no head protection was worn uniformly by Blind Football athletes.

Blind Football remains the sport with the highest incidence of injury across the last three successive Paralympic Games: London 2012, Rio 2016, and Tokyo 2020 [14,16,24,25]. The injury incidence rate in Blind Football in Rio 2016 (22.5 injuries/1000 athlete days) was more than double the overall injury incidence rate at the Rio 2016 Paralympic Games (10.0 injuries/1000 athlete days overall) [14,25]. Although the Rio 2016 and Tokyo 2020 studies did not document the prevalence of head injury as a percentage of overall injury in Blind Football, findings from the London 2012 study demonstrated that head and neck injuries were among the most prevalent, accounting for 25% of all acute injuries in Blind Football [21]. Despite this high prevalence of injury to the head and neck, no concussions were reported in either the London 2012 or the Rio 2016 Paralympic Games [14,24]. As mentioned previously, this was partially because the IPC did not include questions specifically related to concussion in the WEB-IISS until the Rio 2016 Paralympic Games [9]. While nine concussions were reported in the most recent Tokyo 2020 Paralympic Games, experts believe the condition is underreported, particularly in Blind Football [26]. The underreporting of concussions in the para athlete population could reflect the need for increased clinician education regarding concussion recognition and assessment in this population [14]. It is also likely that there is less availability of specially trained medical staff in Blind Football, and there may be an added responsibility on the para athlete to recognize and report concussion symptoms [23]. Ultimately, more education on concussion recognition is needed for all parties involved in Blind Football to improve concussion reporting, management, and clinical outcomes [23].

More recently, Weiler et al. conducted a three-year prospective injury surveillance study in elite English Blind and Cerebral Palsy (CP) Football squads and observed a high proportion (17%) of injuries were to the head and neck in both Blind and CP squads [27]. The data also suggested that, based on hourly exposure, Blind Football carried a higher incidence of injury in both training and matches compared to CP Football [27]. Moreover, a total of 75% of injuries in Blind Football matches were sustained through contact, compared to 50% in CP Football matches [27]. Although these results are expected—considering athletes with VI have limitations to their vision, making it more difficult for them to avoid contact—these results also demonstrate that there is a need for stricter enforcement of the ‘voy’ rule in Blind Football [27]. The ‘voy’ rule was implemented several years ago in Blind Football to limit the amount of contact between players defending and attacking the ball [27]. The rule requires players to shout the word ‘voy’, which translates to “I go” in Spanish, when they are on the defensive to allow the attacking player to determine the positions of the defenders on the field and to avoid a collision with them [27].

To promote the recognition and assessment of suspected concussions in Blind Football athletes, the IBSA recently adopted a new policy known as the ‘Temporary Concussion Substitution’ (TCS) rule [28]. Adapted from the TCS rule first implemented in CP Football, the TCS rule ensures that in the event of a head injury, a substitute player (i.e., a TCS player) can enter the field of play to replace the player with the suspected head injury for a period of 10 min [29]. The rule gives clinicians more time to assess suspected head injuries without the pressure of returning to gameplay and interfering with their expert medical decision [30]. If, after the 10-min concussion assessment, the athlete is deemed safe to return to play by the team medical professional, the TCS player must leave the field of play and be replaced by the player initially withdrawn [28]. However, if the medical professional determines that it is unsafe for the athlete to return to play due to concussion concerns, the substituted player may remain on the pitch as the permanent replacement [28]. By giving clinicians more time to assess suspected cases of concussion, this policy can provide a safer sport for athletes with VI while also allowing the football match to continue uninterrupted with an equal number of players on the field for both teams [31].

Recognizing the high rates of injury to the head, face, and neck in Blind Football, additional rule changes and stricter enforcement of the ‘voy’ rule may be required in the future to limit the amount of contact injuries in the sport [27]. The TCS policy discussed above also demonstrates that safer sports are possible with effective collaborative efforts between sports medicine professionals and sporting organizations such as the IBSA. To create a safer environment for athletes with VI, additional policy and rule changes should continue to be informed by the latest epidemiological research and should also consider the perspectives of all stakeholders involved in the sport (i.e., athletes, coaching staff, medical staff, and event/sport organizers).

### 2.2. Goalball

Goalball is a summer para sport played exclusively by athletes who are blind or vision impaired. Much like Blind Football, players are required to wear opaque eye shades to ensure fair competition [32]. To participate in the sport, athletes must have a VI and be classified as either B1, B2, or B3 (see Introduction for classification criteria). The object of the game is to throw a 1.25 kg ball across the court and into the opposing net to score points [32]. Goalball athletes throw from a standing position and often defend oncoming shots from the opposing team by throwing themselves to the floor to stretch out and intercept the ball in the correct position [32]. There is no designated goalkeeper in Goalball; all three players on each team play both defense and offense simultaneously.

The IPC Medical Committee rated Goalball a ‘3’ out of a possible ‘5’ as the best estimate for concussion risk, putting Goalball at a moderate risk for concussion compared to other para sports [9]. Much like Blind Football athletes, Goalball athletes have no light perception on the field of play and are at risk of colliding with other athletes and objects [9]. In addition, no head protection is worn in Goalball. Lexell et al. found that all sport-related concussions reported among Goalball athletes were related to a collision, and it was hypothesized that the lack of head protection could further increase athletes’ risk of sustaining a concussion in the sport [18].

At the London 2012 Paralympic Games, Goalball had an injury incidence rate of 19.5 injuries/1000 athlete days, the third-highest rate of all sports at the Games [24]. A study by Zwierzchowska and colleagues reported that 44% of athletes who participated in the Goalball European Championship reported an injury of some sort, with 92% of all injuries occurring to athletes’ upper limbs [33]. Gajardo and colleagues, the first group to investigate prior-to-competition injuries and illnesses in Goalball athletes, found an injury prevalence of as high as 64.1% within four weeks prior-to-competition [34]. This finding further emphasizes the value of conducting epidemiological research during regular training and prior-to-competition periods to better understand the burden of injury in VI para sport.

Although all three of the above studies demonstrated a relatively high injury incidence and prevalence in Goalball in general, no concussions were reported. Nevertheless, Lexell et al. found that 31% of all concussions self-reported by para athletes in the SRIIPSS occurred in Goalball [18]. In this study, Goalball athletes sustained the most concussions out of all other sports surveyed [18]. Goalball athletes in this study reported that the mechanism of concussive injury was influenced primarily by their use of eye shades [18].

Given the information we have so far regarding the intrinsic (i.e., athletes with VI) and extrinsic (i.e., use of eye shades) risk factors for concussion sustainment in Goalball, prevention strategies to reduce the risk of concussion in the sport should be implemented. Recently, the IBSA postulated that a combined eye shade with head protection may offer a protective benefit in the head-to-head collisions seen in Blind Football [9]. Results from their internal study on the effectiveness of softshell helmets with integrated eye shades in preventing injuries to the head have yet to be published [23]. Should this type of protective headgear confer a risk reduction benefit in Blind Football, it is worth investigating whether a similar type of headgear could provide a protective benefit in Goalball as well. Impact warning devices should also be considered as an additional preventative measure in Goalball. These devices could be integrated into athlete equipment and objects surrounding the field of play (e.g., goalposts, the ball, barricades, walls, etc.). If an athlete were to come within a certain distance of an object or another athlete, the device could emit a warning sound, alerting the athlete to take a chance to avoid potential head-to-head contact.

### 2.3. Para Judo

Para Judo was first adopted as a Paralympic sport in 1988 and is another sport exclusively for athletes with VI in the Summer Paralympic Games [35,36]. The objective of Para Judo is to throw or takedown your opponent and force them into submission or a pin [35]. Para Judo athletes are subdivided into gender and weight categories in a similar fashion to their non-disabled counterparts in the Olympics [36]. Paralympic judokas are classified into one of two vision classes depending on the degree of their VI: either J1 (i.e., binocular visual acuity ≥2.6 LogMAR) or J2 (i.e., binocular visual acuity between 1.3 and 2.5 LogMAR or a binocular visual field of 60 degrees or less in diameter) [36,37]. The classification system for Para Judo described here came into effect in January 2022. Prior to January 2022, all athletes competed in a single class, regardless of their VI.

Injury incidence data from Olympic-level Judo indicates that judokas are prone to a range of injuries [38]. From the available evidence in para sport, the former also seems to be the case in Para Judo. At the Rio 2016 Paralympic Games, Para Judo had the second highest injury incidence rate out of all other para sports, with an incidence rate of 15.5 injuries/1000 athlete days [14]. Para Judo was once again highlighted as a sport with a high risk for injury at the Tokyo 2020 Paralympic Games. Para Judo’s documented injury incidence rate of 11.6 injuries/1000 athlete days made it the sport with the third highest injury rate and the highest concussion incidence rate (i.e., three concussions total) at the Tokyo Games [16]. These injury rates are consistent with other research in Para Judo.

Fagher et al. investigated the sport-related injury prevalence among Paralympic judokas and found that 84% of athletes suffered from at least one injury during a one-year period [36]. This study also found that most injuries (74%) occurred during Para Judo training. Fagher et al. did not report any concussion injuries in their study [36]. Another study by Kons et al. that measured the prevalence, magnitude, and mechanisms of injury amongst judo athletes with disabilities found that athletes with VI were most affected by injuries compared with athletes with other disabilities (i.e., physical and intellectual impairment) [39]. The main injury mechanism among judo athletes with impairments was found to be associated with direct contact with other athletes in training environments [39]. Interestingly, this study also found a higher prevalence of injuries during routine training for athletes with VI than during competition [39]. Similarly to Fagher et al., Kons et al. reported no concussion injuries [36,39].

Lexell et al. reported that 15% of all concussions in their study occurred in Para Judo [18]. The mechanism of injury for both concussions was related to a collision. Additionally, athletes reported concussion injuries related to being thrown in an unexpected direction [18]. Given the lack of additional data on concussion incidence in Para Judo and the high incidence of injury in general in this sport, more epidemiological concussion injury incidence studies are needed.

In addition to the need for more epidemiological research on concussion in Para Judo, technical research efforts focused on identifying possible risk factors and injury mechanisms in the sport are also needed [36]. Fagher et al. identified that 82% of injuries in Para Judo occurred in tachi-waza, a Judo term that refers to a collection of different throwing techniques [40]. This prompted the authors to suggest that a first step towards injury prevention in general could be to allow matches to continue into ne-waza (i.e., a Judo term that refers to a collection of ground techniques) and minimize time spent in tachi-waza [36,41]. In the future, more studies should assess different techniques, throws, and falls in Para Judo to determine whether any other techniques may be putting judokas at an increased risk of sustaining a concussion [36]. It was also suggested by the same authors that video footage analysis could be used in conjunction with injury surveillance data to better understand additional injury mechanisms at play in Para Judo [36]. from the authors of both the Rio 2016 injury incidence study and the Fagher et al. study agree that more research is needed to identify injury mechanisms and risk factors compounding the risk of concussion in Para Judo [14,36]. In turn, this research will allow for the development of effective strategies for injury prevention in this sport [14,36].

### 2.4. Para Cycling

Para Cycling is a multi-disability sport, featuring events for athletes with VI and athletes with physical impairment [42,43]. There are four main types of cycles used in Para Cycling, based on impairment, and they include handcycles, tricycles, standard upright bicycles, and tandem cycles [43]. Para cyclists competing on standard bicycles and tandem cycles may compete in both road and track events, while para athletes competing on handcycles and tricycles may only compete in road events [43]. Tandem cyclists are visually impaired, and they ride with a sighted guide in front known as a “pilot” [42,43]. Para athletes with VI are designated B1, B2, or B3 in accordance with the IBSA classification standards [43]. All tandem cyclists with VI compete together under a single classification (i.e., B) in the same events [43].

While head protection is worn during competition and para athletes with VI have a sighted pilot steering the tandem bike on the road/track, the high speeds that can be achieved in this sport predispose athletes to harmful falls [9,44]. For this reason, the IPC Medical Committee rated para road cycling a ‘5’ out of a possible ‘5’ as the best estimate for concussion risk, making para road cycling one of only two para sports with the highest perceived concussion risk [9]. According to Clarsen and colleagues, sport-related concussions remain a poorly quantified injury in both non-disabled sports and para sport cycling [45].

Injury surveillance data from the London 2012 Paralympic Games indicated that both para track cycling and para road cycling had some of the highest proportions of acute traumatic injuries out of all para sports at the 2012 Games [24]. Track cycling saw 75% of total injuries classified as acute, and similarly, road cycling saw 71% of total injuries classified as acute [24]. While the study reported injuries by anatomical region, the data was not sport-specific, so it is unknown how many of the acute injuries seen in Para Cycling were to the head or face region [24]. As for the Rio 2016 Paralympic Games, Para Cycling (both track and road) had a moderate injury incidence rate compared to other sports of 7.0 injuries/1000 athlete days [14]. Although injuries by anatomical area were reported in the study, these results were once again not sport-specific, making it difficult to know how many injuries occurred to the head region in Para Cycling [14]. Unfortunately, Rio 2016 also observed the first fatal injury of an athlete in a Games setting; a para cyclist with a physical impairment suffered a head injury during competition and later succumbed to their injury [14]. This catastrophic event highlights that life-threatening head injuries occur in Para Cycling, and we must continue to understand the risk factors and injury mechanisms involved to prevent these injuries from happening.

Tokyo 2020 was the first Paralympic Games to report documented cases of concussion in the sport to date [16]. Two concussions were reported to have taken place in Para Road Cycling; however, the impairment types of the athletes that sustained the concussion injuries are unknown [16]. Future injury reporting should consider publishing concussion injury incidence by impairment type, so conclusions related to injury mechanisms and risk factors for injury can be drawn.

Clarsen and colleagues highlighted that para cyclists have received little research attention to date [45]. Outside of studies conducted at the Paralympic Games, only one epidemiological study has focused on understanding the characteristics of sport injuries sustained by para cyclists, specifically [44]. Going forward, researchers should strive to conduct high-quality epidemiological studies following the principles laid out in the most recent consensus statement on the methods for epidemiological studies in competitive cycling [45]. Some of these recommendations include reporting cycling-specific collision agents (i.e., whether the collision was with a person, inanimate object, vehicle, etc.), reporting collision mechanisms (i.e., equipment failure, avoiding objects, surface quality, etc.), and reporting related circumstances such as environmental factors (i.e., wind, rain, temperature, etc.) and track surface conditions [45]. Improved documentation of injury mechanisms in Para Cycling will help guide key stakeholders (i.e., governing bodies, race organizers, and equipment manufacturers) in the development of successful injury prevention initiatives [45].

### 2.5. Para Swimming

Para Swimming has been featured in every Summer Paralympic Games program and sees some of the highest numbers of para athlete participants [14,46]. It is also a multi-disability sport where athletes of all eligible impairment types can compete in a range of impairment classifications [46,47]. Athletes with VI are categorized into one of three sport classes: S/SB11, S/SB12, and S/SB13, which are equivalent to the B1, B2, and B3 classifications, respectively [47,48]. Athletes in the S/SB11 sport class, except those with prosthetic eyes, are required to wear blackened goggles to standardize the light perception amongst competitors in this category and to ensure fair competition [47,48]. S/SB11 swimmers are also required to have an assistant (i.e., known as a “tapper”) at both ends of the pool to “tap” the athlete to let them know they are approaching the pool end wall [48]. Swimmers in the S/SB12 and S/SB13 sport classes may choose to use a tapper if they wish [48].

Injury incidence results from the Rio 2016 and Tokyo 2020 Paralympic Games found that Para Swimming had a low-to-moderate injury incidence rate compared with other summer sports (i.e., 7.1 injuries/1000 athlete days and 3.1 injuries/1000 athlete days, respectively) [9,14,16]. Thus far, one concussion was reported in Para Swimming at the most recent Tokyo 2020 Paralympic Games [16]. Magno e Silva and colleagues found that overuse injuries made up 80% of all injuries reported in a group of 18 Brazilian Para Swimming athletes, with acute injuries making up the remaining 20% [47]. A notable observation from the Magno e Silva et al. study was that there were no reported injuries to the head or face region [47]. This result suggests that the use of a tapper in Para Swimming may be an effective method of preventing acute traumatic injuries to the head or face, such as concussion injuries [47].

In contrast, Lexell et al. found that 15% of concussions reported among Swedish para athletes over a period of one year occurred in Para Swimming [18]. Both concussions were sustained in Para Swimming athletes with VI [18]. The mechanism of injury was reported to have been caused by swimming into the wall, and the athletes reported that they believed their VI influenced the injury [18]. Athletes with VI reported that had there been more tappers available and better knowledge and awareness of the dangers of concussion amongst coaches, their sport-related concussions could possibly have been prevented [18]. Even though Para Swimming may be associated with a lower incidence of injury in general compared to other para sports [14], concussions can still occur, and preventative measures should still be investigated and prioritized in this sport as well.

### 2.6. Para Alpine Skiing

Para Alpine Skiing is a downhill racing sport adapted for athletes with disabilities and is one of only two Winter Paralympic sports for athletes with VI [49]. Athletes with VI are classified as either B1, B2, or B3, depending on their level of VI [49]. In addition, athletes with VI ski with the assistance of a guide [49]. The guide skis in front of the athlete and gives verbal direction to the athlete through a radio-frequency headset and microphone [49].

Para Alpine skiers with VI compete in four skiing disciplines: slalom, giant slalom, Super G, and downhill. Across these disciplines, athletes can reach speeds of up to 116 km/h [49]. As a result of the high speeds and potential for impact in this sport, injuries are commonplace [49]. It is no surprise then that the IPC Medical Committee rated downhill Para Alpine for athletes with VI a ‘5’ out of a possible ‘5’ as the best estimate for concussion risk in this sport [9]. Downhill Para Alpine, in particular, is one of two sports with the highest perceived concussion risk out of all other para sports, the other being Para Cycling (Road) [9,49].

There is anecdotal evidence that concussions and other head and neck injuries are on the rise and are approaching the injury incidence of the more common injuries (e.g., shoulder joint and knee ligament injuries) in both standing and seated Para Alpine athletes [49,50]. At the Sochi 2014 Paralympic Winter Games, 31 injuries to the head, face, and neck were reported, which accounted for an injury incidence rate of 4.7 injuries/1000 athlete days [50]. The incidence rate of all injuries in Para Alpine was found to be 41.1 injuries/1000 athlete days, significantly higher compared to all other winter sports at the Sochi 2014 Games [50]. The incidence of injury recorded at the Sochi 2014 Paralympic Games was three times the incidence of injury recorded at the Sochi 2014 Olympic Winter Games, suggesting a higher risk of injury in athletes with impairment compared to their non-disabled counterparts [9,50]. Even so, no concussions were reported in the Sochi 2014 Paralympics study, again because the WEB-IISS was not specific enough to capture concussion injury data at that time [50]. At the Pyeongchang 2018 Paralympic Winter Games, 4 of the 29 injuries reported to the head, face, or neck region were concussions, two of which occurred in Para Alpine [15]. There was no further indication as to whether these concussions occurred in athletes with VI or in athletes with a physical impairment.

The high injury incidence rate in Para Alpine at the Sochi 2014 Paralympic Games prompted medical and sport-technical experts to create an action plan to reduce the risk of injury for the following Pyeongchang 2018 Games [23]. It was hypothesized that modifiable factors (i.e., course design, number of training runs, the command and control structure between the technical and medical staff, etc.) and environmental factors (i.e., temperature and altitude of the skiing venues in Sochi and their effects on snow conditions) were the likely contributors to the high injury rate observed in Sochi 2014 [51]. In light of this information, the IPC Medical Committee engaged in discussions with the World Para Alpine Skiing management team to ensure injury risk and prevention were top priorities for Pyeongchang [51]. Measures for Pyeongchang included: more training runs; earlier start times; a more optimal start location on the course; the widening of the course; official pre-Games technical and medical briefings; and the appointment of an independent race director who would facilitate the investigation into safety issues and have the final call to amend, postpone, or cancel an event if the conditions were deemed to be too hazardous [51]. The outcomes of these measures appear to have been successful in lowering the total number of injuries (i.e., 39 injuries in Pyeongchang compared to 98 in Sochi), decreasing the number of acute injuries (i.e., 17 acute injuries in Pyeongchang compared to 48 acute injuries in Sochi), and decreasing the number of injuries in downhill skiing from 21 in Sochi to only 5 in Pyeongchang [52].

The above is an example of how effective collaborations between sports medicine and sport management can realize changes in the policy, rules, and/or laws of sport to enhance the safety of athletes [23,52]. In terms of reducing the risk of concussion in Para Alpine, similar approaches can and should be taken going forward. Recently, World Para Alpine Skiing, World Para Snowboard, and World Para Nordic Skiing introduced concussion-specific rules to permit non-medically trained personnel to initiate a primary assessment of concussion, in accordance with Para Alpine skiing protocols [23]. Going forward, conscious efforts should be made to include sports medicine professionals when informing sport-technical policy to ensure the safety of athletes, which remains a top priority amongst sports organizations [52].

## 3. Challenges and Limitations of Current Epidemiological Research

Injury and illness epidemiological research in para sports in general has improved over the last decade. This is largely due to the implementation of the WEB-IISS at every Summer and Winter Paralympic Games since 2012, the SRIIPSS, and the recent sport-specific and impairment-specific injury epidemiological studies. An understanding of common injury mechanisms is starting to emerge in specific para sports. However, our understanding of the risk factors and mechanisms of concussive injury in para sports is preliminary at best due to the limited number of studies that have been undertaken.

A major limitation of current research is that most epidemiological studies are conducted during major competitions such as the Paralympic Games and other International World Championships. It should be noted that during the period of the Paralympic Games and other major international competitions, para athletes are typically training and competing less frequently than during the normal season [18]. The prospective cohort study by Lexell et al., which looked at concussion injuries over a period of 52 weeks, found that most sport-related concussions (69%) occurred during sport-specific training, with a minority of concussions (31%) taking place during competition [18]. Likewise, another study by Derman et al. found that pre-competition injuries in general were significantly higher compared to injuries during the competition period itself at the Rio 2016 Paralympic Games [14]. Consequently, only assessing concussion injury risk during major competitions is likely to vastly underestimate the true incidence of concussion in para sports. More studies are needed that prospectively study concussion in para sports over longer periods of time (i.e., during both training and competition).

Injury surveillance systems should ask specific questions related to concussion and report concussion-related injuries with more specificity, such as documenting the mechanism of injury, athlete impairment type, and athlete classification within each impairment type. As mentioned earlier, questions specifically related to concussion were not included in the WEB-IISS system until the Rio 2016 Games [9]. As a result, both the London 2012 and the Sochi 2014 injury surveillance studies failed to report any confirmed cases of concussion. The Tokyo 2020 Paralympic Games injury surveillance study was significant in its contribution to the understanding of concussion incidence in para sports [26]. The study included a supplemental file where an in-depth description of all nine concussion injuries was provided [16]. The supplemental file included information on the reporting of the concussion injuries (i.e., whether the athlete had been concussed before, whether the SCAT5 was used in the assessment of the concussion, etc.) and the nature of the concussion injuries (i.e., sport-related/non-sport-related, athlete impairment type, main presenting symptoms, estimated time lost to injury, etc.) [16,26]. Where possible in future studies, it would be beneficial to conduct multivariate analyses to allow for the identification of risk factors and mechanisms of injury specific to athlete impairment type and sport. When documenting mechanisms of injury, athlete impairment type, and classification information, care should be taken to ensure the privacy of athletes so that they cannot be identified directly from the information made available.

Lexell and colleagues’ longitudinal prospective cohort study was well designed in that it measured concussion injury incidence during both training and competition periods; however, there are some limitations to the study. For instance, the study included no athletes from the high-risk summer sport of Blind Football and only two athletes from the high-risk winter sport of Para Alpine [18]. The incidence of concussion injuries in para sports may have been even greater had more athletes from high-risk sports been included [18]. Additionally, only athletes from Sweden were invited to participate in this study [18]. It is possible that concussion injuries could occur differently in other nations due to differences in training approaches, medical support, and/or concussion education programs. Future prospective cohort studies with athlete populations from different nations and across different continents are needed. Furthermore, 107 Swedish para athletes agreed to participate in the study out of a total of 150 athletes (71% of all athletes in the Swedish Paralympic Program) [18,53]. While the level of engagement of Swedish para athletes in the study is impressive, a larger sample size would have enabled the investigators to make more substantive claims regarding concussion injury risk, the mechanisms of injury, and the intrinsic and extrinsic risk factors contributing to concussion sustainment in para sport [18]. Considering a total of 4403 para athletes competed in the Tokyo 2020 Paralympic Games [54], a larger sample size is attainable with the creation of a multi-national prospective cohort study to measure the incidence of sport-related concussion.

Lack of education among athletes, coaches, sports organizations, and even medical professionals regarding concussion recognition is a factor contributing to the underreporting of sport-related concussions in para sport [9,18]. A study that surveyed para athletes who participated in Wheelchair Basketball found that as many as 44% of athletes who experienced a sport-related concussion did not report it [55]. Among athletes surveyed, the three most common reasons for failing to report a concussion consisted of: firstly, not wanting to be removed from the game; secondly, not thinking the injury was serious enough; and lastly, not knowing whether it was indeed a concussion [55]. Another study found that athletes with VI who had suffered a concussion believed that better knowledge and awareness among their coaching staff could have possibly prevented their concussion [18]. Finally, during the Rio 2016 Paralympics, no concussions were reported among team medical staff, despite several incidents where athletes were observed to suffer a blow to the head followed by an unsteady gait [14]. These findings demonstrate the need for concussion education as an additional critical supporting structure to limit the prevalence of concussion injuries and create safer competition for all para athletes. Moreover, concussion education should be in a format accessible to athletes with impairments. Accessible concussion education for athletes with VI can be achieved by providing resources in braille, using larger font sizes and greater contrast, and ensuring online resources can be equipped with text-to-speech functionality [23]. As more research becomes available in terms of how athletes with VI experience concussions, concussion education should include unique considerations to keep in mind when an athlete with VI experiences a concussion.

## 4. Recommendations for Future Research

Future research aiming to close the knowledge gap on concussion incidence data in para sports should consider the following recommendations in their study design: Firstly, epidemiological studies should measure concussion incidence and injury risk prospectively during athlete training periods in addition to during major competitions. Several studies to date have found a higher incidence of concussion injury and injury in general during sport-specific training as opposed to during competitive periods [14,18,27,34,36,39]. Additionally, for every concussion injury reported, injury surveillance studies should consider documenting the mechanism of injury, athlete impairment type, athlete classification type, and the para sport to promote the identification of risk factors and the creation of sport-specific injury prevention measures. When publishing this data, measures should be taken to ensure the identity of athletes remains confidential. Lastly, concussion injury incidence data is urgently needed in recreational sport settings as well as in athlete populations based in Africa, Asia, and Oceania. The majority of published research thus far has been concentrated in elite sport settings only and has investigated athlete populations primarily located in the Americas or Europe.

Researchers interested in creating safer sporting environments for para athletes should consider the following recommendations for future studies, based on the available evidence from sport-specific concussion incidence data presented earlier. Studies investigating the effectiveness of protective headgear in limiting concussion injuries in para sports such as Blind Football and Goalball are needed to inform future protective equipment recommendations. Research studies investigating the effectiveness of new sport rules and policy changes on the incidence of concussion injuries are also warranted to inform future policies meant to protect athletes from concussion injuries. The inclusion of a temporary concussion substitution policy in Blind Football and the changes made to the course design in Para Alpine Skiing at the Pyeongchang 2018 Paralympic Winter Games are examples of how collaborations between researchers, sports medicine, and sport management professionals can enhance the safety of athletes [29,52]. Finally, research studies are needed to investigate current concussion knowledge and awareness amongst athletes, coaches, health care providers, and sport administrators in para sport to understand for whom and where concussion education efforts should be directed.

## 5. Conclusions

In recent years, we have seen injury epidemiology research in para sports improve tremendously. While concussion injury incidence across specific para sports is beginning to be identified, knowledge gaps concerning the actual burden of concussion injury in para athletes with VI persist.

This review examined the available evidence of concussion incidence rates across six different para sports for athletes with VI and found a disproportionate burden of concussion in para athletes with VI when compared to other para sports. Modifying sport rules, introducing protective equipment, and incorporating additional safety measures during play may be effective solutions for reducing the prevalence of concussion injuries in athletes with VI. The review highlights the urgent need for objective concussion incidence measures to determine VI athlete concussion injury rates and risks more accurately. More prospective, sport-specific concussion incidence studies are needed to understand why athletes with VI are at a seemingly higher risk for concussion injuries than athletes with other impairment types and non-disabled athletes. Research examining the mechanisms of concussion injuries in athletes with VI in both training and competitive environments is needed. In addition, research investigating the risk factors for concussion injuries in both elite and recreational sports settings is warranted. The continued collection of concussion incidence data in VI para sport settings is crucial to the creation of effective and targeted preventative measures to reduce the risk of concussion in VI para sports.

## Data Availability

No new data were created or analyzed in this study. Data sharing is not applicable to this article.

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
