# Peer review of "An Urgent Call for Concussion Incidence Measures in Para Sport for Athletes with Vision Impairment: A Narrative Review"

_healthcare, 2024, doi:10.3390/healthcare12050525_

Round 1

Reviewer 1 Report

Comments and Suggestions for Authors

This is a very interesting narrative review on concussion incidence and risk in para athletes with visual impairment (VI). Overall, the review was well-written and well-organized. It serves as a clear call-to-action to better describe concussion incidence and to develop and test preventative strategies for athletes with VI. In the Introduction, there are a few minor issues that should be addressed prior to publication. Please see below for specific suggestions.

 Minor Comments:

 Throughout the manuscript, please be consistent with either spelling out visual impairment or using the VI acronym.

 Page 1, Line 36: Consider adding a clause which indicates how often people experience lasting (i.e., non-transient) deficits/issues from concussion with a relevant citation.

 Page 1, Lines 36-38: Definition of sports-related concussion feels redundant with previous definition of traumatic brain injury. Consider integrating these definitions.

 Page 2, lines 68 – 75. It appears that the authors are providing two examples of concussion incidence in athletes with VI. The first example is 4 of 9 athletes from the Tokyo 2020 games, and the second example references approximately 56% of athletes from an unnamed event. As written, it appears that the sentence describing an ~56% aligns with the previous sentence that includes a 4 of 9 statistic, but the percentage does not match (as 4/9 is ~44%). Because the authors are trying to claim that para athletes with VI are possibly more susceptible to concussion, these two examples need to be explained much more clearly. If there are not two distinct examples, but only one example, then the authors should soften their language around prevalence of VI in Paralympic events OR merge this statistic with the next paragraphs that show empirical evidence of higher concussion incidence in VI.

 Page 3, Line 103: Considering adding ‘adequate data’ as there are some data – there does not appear to be a complete absence of incidence data by impairment type and sport, as the preceding paragraph described those type of data.

 Page 3, Line 111: Is the “VI” qualifier before para sports accurate or is it more accurate to simply say “para sports.” In other words, are these sports higher risk for all athletes or just those with VI – if the latter than the opening sentence of this paragraph seems contradictory to this final sentence.

 Page 3, Line 112 – Depending on how the preceding paragraph is concluded, this paragraph may need a transition sentence to move from all para sports to VI para sports. Additionally, it may help the reader to know – prior to Section 2. Sport-Specific Concussion Incidence – that some para sports are exclusively played by athletes with VI whereas other sports include athletes with VI and other types of impairments.

Reviewer 2 Report

Comments and Suggestions for Authors

First of all, I would like to thank the editorial team for the opportunity to review this article.

Changes need to be made to certain parts of the article before it can be accepted for publication.

Abstract: although it shows the need for the study, more information is needed on the specific aim of the study and on the methods used to carry out the review. 

Introduction: lines 31 and 32. It seems that only people with impairments benefit from team sports. I would redefine this sentence and include concrete examples of the improvements obtained. 

Since during the introduction the importance of visual impairment is shown, what is considered as visual impairment, what types are there? Are there differences in concussions according to visual impairment type or degree? This should be explained. 

Check that visual impairment appears as VI after the first time it has appeared (example: line 114). 

2. Sport-Specific Concussion Incidence is very well defined. I congratulate the authors who have made a great description of each of the sports, the types of VI in each and the risks involved.

During the article, some practical applications as well as future lines of research are shown. In order to give the reader the possibility to extract information and data from your article, I recommend you to make a specific section in which this information is included (even if it is a recapitulation of what was previously mentioned in other parts of the article). I strongly recommend that you include this section between point 3 and the conclusion. 

I would include a specific aim in the introduction section, and I would answer it in the conclusions. I think it would be easier for the reader to know what is intended and what is found in this regard. 

Despite these minor modifications, I would like to congratulate the authors for this great work. It reports current news on the risk of concussion in VI and provides information of great relevance in the scientific and practical fields. If the requested modifications are carried out the article can be accepted. 

Comments on the Quality of English Language

A grammatical revision of the manuscript is needed. I believe there are errors that hinder comprehension and should be corrected. 

Reviewer 3 Report

Comments and Suggestions for Authors

Dear Authors,

You have written an interesting Narrative Review paper focusing on understanding and addressing the risks of concussion injuries to ensure the safety and well-being of all para-athletes.

The introduction describes the problematic area well with up-to-date research. However, it does not discuss the vast terminology used for athletes with disabilities as various terms are used: para, adapted, disabilities, inclusive, special needs, etc. The paper from the case of judo might help the authors in highlighting this problem:

https://archbudo.com/view/abstract/id/13347

In the sport of judo a study by Kons et al. (2022) might help in additional injury mechanism description.

https://doi.org/10.1123/jsr.2021-0352

From the sport of para Taekwondo, an important study that fits this review was published on Injury Incidence and Severity at the 5th World Para-taekwondo Championships. This sport might also be included to add to the comprehensiveness of this review.

https://doi.org/10.15758/jkak.2017.19.1.45

Additionally, the Wheelchair rugby concussion area as a paralympic sport has also not been identified: several studies were done on this problem:

https://link.springer.com/book/10.1007/978-3-030-83004-5

DOI: 10.1097/PHM.0000000000001630

https://www.ncbi.nlm.nih.gov/pmc/articles/PMC9679180/

These sports should be additionally analysed.

Future guidelines are well identified and stated to increase the quality of future studies.

Overall the review is solid. However, some additional literature should be included. Therefore, I recommend a major revision.

Kind regards

Comments on the Quality of English Language

Minor editing of the English language required

Round 2

Reviewer 3 Report

Comments and Suggestions for Authors

Dear Authors

Thank you for addressing all of my questions and suggestions adequately. The manuscript's quality and clarity have improved.

Therefore, I recommend acceptance in its current form

Congratulations

Comments on the Quality of English Language

Minor editing of the English language required